# Unstable Unlearning: The Hidden Risk of Concept Resurgence in Diffusion Models

## Abstract

Text-to-image diffusion models rely on massive, web-scale datasets. Training them from scratch is computationally expensive, and as a result, developers often prefer to make incremental updates to existing models. These updates often compose fine-tuning steps (to learn new concepts or improve model performance) with "unlearning" steps (to "forget" existing concepts, such as copyrighted data or the ability to generate explicit content). In this work, we demonstrate a critical and previously unknown vulnerability that arises in this paradigm: even under benign, non-adversarial conditions, fine-tuning a text-to-image diffusion model on seemingly unrelated images can cause it to "relearn" concepts that were previously "unlearned." We comprehensively investigate the causes and scope of this phenomenon, which we term *concept resurgence*, by performing a series of experiments based on fine-tuning Stable Diffusion v1.4 alongside "mass concept erasure," the current state of the art for unlearning in text-to-image diffusion models (Lu et al., 2024). Our findings underscore the fragility of composing incremental model updates, and raise new serious concerns about current approaches to ensuring the safety and alignment of text-to-image diffusion models.

## 1 Introduction

Modern generative models are not static. In an ideal world, developing new models would require minimal resources, allowing users to tailor unique, freshly trained models to every downstream use case. In practice, making incremental updates to existing models is far more cost-effective, which is why it is common for models developed for one context to be updated for use in another. This paradigm of updating pre-trained models is widely considered beneficial, as it promotes broader and more accessible development of AI. However, for sequential updates to become a sustainable standard, it is critical to ensure that these updates compose in predictable ways.

Developers commonly update models to acquire new information or improve performance—for example, by fine-tuning an existing model on a novel dataset tailored to a particular use case. But sometimes, developers also seek to *remove* information from an existing model. One prominent example is *machine unlearning*, which aims to efficiently update a trained model to "forget" portions of its training data (Cao & Yang, 2015; Nguyen et al., 2022b; Belrose et al., 2024), in order to respond to privacy concerns. This is especially important for compliance with regulations like the General Data Protection Regulation (GDPR) "right to be forgotten" (European Parliament and Council of the European Union, 2016).

Here, we focus on the related notion of "concept unlearning" in the context of text-to-image diffusion models (hereafter, referred to as "diffusion models"). In contrast to machine unlearning, which targets particular data points, concept unlearning seeks to erase general categories of content, such as offensive or explicit images. There has been substantial recent progress in this area. For example, the current state-of-the-art in "mass concept erasure" (MACE) can now effectively erase dozens of concepts from a pre-trained diffusion model (Lu et al., 2024). This is useful when dealing with generative models where undesired concepts are unknown during pre-training, but need to be erased as the model is adapted for different users or applications.

Our work begins with a surprising observation: **fine-tuning a diffusion model can re-introduce previously erased concepts**, even when fine-tuning on unrelated concepts (see Figure 1 for an example). This hidden vulnerability, which we call *concept resurgence*, challenges the current paradigm

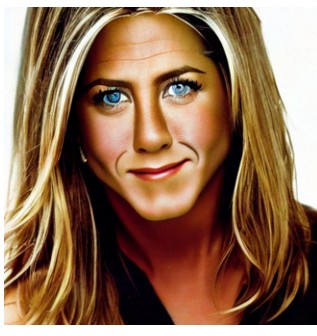 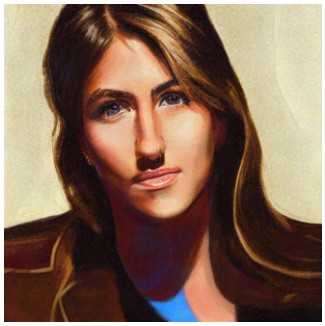 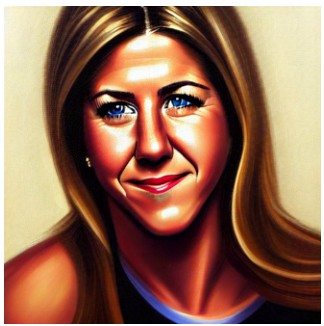

(a) Stable Diffusion v1.4           (b) MACE           (c) Additional Fine-tuning

Figure 1: Images generated by the prompt "A portrait of Jennifer Aniston." Stable Diffusion v1.4 successfully generates this image (a), and Mass Concept Erasure (MACE) successfully induces the pretrained model to "forget" this concept (b). However, subsequent fine-tuning on an unrelated set of randomly selected celebrity images reintroduces the ability to generate the target concept (c).

of composing model updates. A developer who seeks to protect downstream users from undesirable concepts (e.g., harmful content or copyright infringement) cannot guarantee that concept unlearning will prevent their accidental reintroduction. Similarly, a consumer who fine-tunes a "safe" model might inadvertently reintroduce undesirable content.

This paper systematically explores concept resurgence, identifying it as a critical and previously unrecognized vulnerability in diffusion models. Our primary contributions are as follows:

- **The prevalence of concept resurgence.** Through a series of systematic experiments, we investigate the conditions under which concept resurgence occurs. Our results reveal that this phenomenon does not require that the model is fine-tuned on data which is similar to the unlearned concept(s), or that the fine-tuning set is chosen adversarially to "jailbreak" the model. Instead, we show that concept resurgence can occur under common and benign usage patterns. Even well-meaning engineers may unintentionally expose users to unsafe or unwanted content. Figure 1 presents a striking yet representative example of this phenomenon.

- **The causes of concept resurgence.** We conduct a thorough examination of both the unlearning algorithms and the fine-tuning data involved in concept resurgence. We quantify the extent of the problem across a range of standard benchmarks, finding that the degree of concept resurgence is closely related to the choice of mapping concept (i.e., the more generic / unrelated the concept, the less resurgence) and degree of regularization imposed during unlearning.

**Organization.** The remainder of this paper is organized as follows. Section 2 is dedicated to background and related work. In Section 3, we quantify the extent of concept resurgence across a variety of standard benchmarks. We investigate the influence of unlearning and fine-tuning algorithms in Section 4, followed by an analysis of the role of fine-tuning data in Section 5. Finally, Section 6 discusses limitations and outlines directions for future research.

## 2 BACKGROUND AND RELATED WORK

**Machine unlearning.** We build on a growing literature on *machine unlearning* (Bourtoule et al., 2021; Nguyen et al., 2022a; Kurmanji et al., 2023; Cao & Yang, 2015; Gupta et al., 2021; Suriyakumar & Wilson, 2022; Sekhari et al., 2021; Ghazi et al., 2023; Kurmanji et al., 2023; Lev & Wilson, 2024), which develops methods for efficiently modifying a trained machine learning model to *forget* some portion of its training data. In the context of classical discriminative models, machine unlearning is often motivated by a desire to preserve the privacy of individuals who may appear in the training data. A key catalyst for this work was the introduction of Article 17 of the European Union General Data Protection Regulation (GDPR), which preserves an individual's "right to be forgotten" (European Parliament and Council of the European Union, 2016). More recent work in machine unlearning has expanded to include modern generative AI models, which may reproduce copyrighted

material, generate offensive or explicit content, or leak sensitive information which appears in their training data (Zhang et al., 2023a; Carlini et al., 2023). Our work focuses specifically on unlearning in the context of text-to-image diffusion models (Ho et al., 2020; Rombach et al., 2021). The literature on diffusion models has grown rapidly over the last few years; though we cannot provide a comprehensive overview here, we refer to Zhang et al. (2023a) for an excellent recent survey.

**Concept unlearning.** Our work is directly inspired by a line of recent research that proposes methods for inducing models to forget abstract *concepts* (Belrose et al., 2024; Lu et al., 2024; Fuchi & Takagi, 2024; Gandikota et al., 2024), as opposed to simply unlearning specific training examples. A key challenge in this context is maintaining acceptable model performance on concepts that are not targeted for unlearning, especially those closely related to the erased concepts. At the time of this work, Lu et al. (2024) (MACE) is the state of the art in terms of both erasure performance and image generation quality after unlearning.

**MACE: mass concept erasure in diffusion models.** We build directly on the recent work of Lu et al. (2024) for mass concept erasure (MACE) in diffusion models, which establishes the state-of-the-art in concept unlearning for diffusion models. At a high level, MACE fine-tunes a model to erase certain target phrase (e.g., "an image of a ship") and their related concepts (e.g., "an image of a boat") by using a combination of cross-attention refinement and low rank adaptation (LoRA) (Hu et al., 2021). Cross-attention refinement modifies the "key" embeddings associated with each token co-existing in the target phrase with the corresponding "key" embedding of co-existing words in a more generic phrase. The second step, LoRA fine-tuning, perturbs the weights of the model to minimize activations in regions which correspond to the target phrase; these regions are identified by segmenting the image using Grounded-SAM (Kirillov et al., 2023; Liu et al., 2023). These perturbations are learned via low rank adapatation (LoRA) of the model parameters (Hu et al., 2021). Finally, the LoRA modules corresponding to each erased concept are combined to produce a final model by formulating the "fusion" of multiple LoRA modules as a quadratic programming problem. For additional detail on MACE we refer the reader to Lu et al. (2024).

**Attacking machine unlearning systems.** Finally, a recent line of research explores data poisoning attacks targeting machine unlearning systems, including Chen et al. (2021); Marchant et al. (2022); Carlini et al. (2022); Di et al. (2023); Qian et al. (2023); Liu et al. (2024). These works show that certain new risks, such as camouflaged data poisoning attacks and backdoor attacks, can be implemented via the "updatability" functionality in machine unlearning, even when the underlying algorithm unlearns perfectly (i.e., simulates retraining-from-scratch). In contrast, our work exposes a qualitatively new kind of vulnerability in machine unlearning, where a previously forgotten concept may be reacquired as a consequence of *additional* learning.

## 3 Composing Updates Causes Concept Resurgence

As discussed in Section 1, the scale of modern diffusion models has motivated a new paradigm in which updates to pretrained models are incrementally composed to avoid retraining models from scratch. These updates broadly take the form of one of two interventions: either the model is updated to learn a new concept, or it is updated to "unlearn" an unwanted concept. The standard procedure for learning new concepts is to curate a dataset of images representing the new concept of interest and fine-tune the model on this dataset. Similarly, to unlearn an unwanted concept(s), an "unlearning" algorithm will typically update the weights of the pretrained model in an attempt to ensure that the model no longer generates content associated with that concept. These two steps may be repeatedly composed over the lifetime of a deployed model. This paradigm raises an important question:

*To what extent is concept erasure robust to compositional updates?*

We begin our investigation with Stable Diffusion v1.4, and apply MACE to a pair of unlearning tasks: celebrity erasure and object erasure. We describe these tasks in detail below. For each task, we fine-tune the model on a random set of in-domain concepts after MACE has been applied. For example, in the context of celebrity erasure — where the goal of the erasure task is to "unlearn" the ability to generate images of a particular celebrity — we further fine-tune the resulting model on a random set of celebrity images (which exclude the unlearned celebrity). This is intended to simulate the real world paradigm of composing unlearning with unrelated fine-tuning steps, the latter of which are intended to help the model learn new concepts or otherwise improve performance. In particular,

we do not fine-tune the model on adversarially chosen concepts, as our goal is to understand whether *benign* updates can degrade or otherwise alter performance. For work on adversarial attacks and/or jailbreaking of text-to-image diffusion models, see Ma et al. (2024); Yang et al. (2024); Dong et al. (2024).

Via these experiments, we uncover a phenomenon we term *concept resurgence*: composing unlearning and fine-tuning may cause a model to regain knowledge of previously erased concepts. Below we provide additional detail on each of these tasks and quantify the degree of concept resurgence.

**Celebrity Erasure.** Following Lu et al. (2024), the first benchmark we consider is inducing the model to forget how to generate certain celebrities (the "erase set") while retaining the ability to generate others (the "retain set"). We benchmark Stable Diffusion v1.4 on four subtasks in which we apply MACE to unlearn 1, 5, 10 or 100 celebrities, and then evaluate whether the model succeeds in generating images of these celebrities (e.g., after being prompted with "A portrait of [erased celebrity name]"). To ensure consistency, both the subtasks and prompts are identical to those in Lu et al. (2024); the full set of celebrities in each subtask, along with the prompts use to evaluate the model, are provided in Appendix C.1. We quantify model performance by separately computing the mean top-1 accuracy of the Giphy Celebrity Detector (GCD) (Hasty et al., 2019) on both erased and retained celebrities.[1] Additionally, we quantify the model's ability to continue to generate general concepts using the CLIP Score (i.e. the cosine similarity between the prompts and generated images) computed on a random sample of 5000 captions from MSCOCO (Lin et al., 2014) (COCO-5K) and the FID between the generated images from pretrained Stable Diffusion and those from our experiments for COCO-5K.

**Object Erasure.** Following Lu et al. (2024), the second benchmark we consider is inducing the model to forget how to generate certain types of objects from the CIFAR10 dataset (the "erase set") while retaining the ability to generate others (the "retain set"). We apply MACE to Stable Diffusion v1.4 across four subtasks, where we unlearn individual objects (automobile, ship, bird) as well as a set of five objects (automobile, ship, bird, cat, and truck). Next, we evaluate whether the model can generate images of these objects and their synonyms (e.g., after being prompted with "a photo of the [erased object]"). Both the full set of erased objects and retained objects, along with the prompts used to evaluate the model, are provided in Appendix C.1. As in the celebrity erasure task, we adopt the set of concepts to be erased, evaluation prompts and other hyperparameters from Lu et al. (2024).[2] We quantify model performance by computing the CLIP accuracy on the set of evaluation prompts. Following Lu et al. (2024), we do not compute FID-5K and CLIP-5K for the object erasure task; COCO-5K is itself composed of common objects, and the goal of this evaluation is to assess performance on generic concepts unrelated to the erasure task.

As discussed above, object erasure and celebrity erasure are two of the four tasks considered in Lu et al. (2024). We discuss our choice to exclude the other two (artistic style and explicit content erasure) in Appendix A.

**Evaluating concept resurgence.** We present representative examples to characterize the degree of "typical" concept resurgence in Figure 2 and Figure 3, and curate specific examples of this vulnerability in Figure 4.[3]

As Figure 2 demonstrates, concept resurgence can occur *in degrees*, as some concepts are not reintroduced at all (e.g., Melania Trump), and others are only partially reintroduced (e.g., Barack Obama). Furthermore, Figure 3 demonstrates that concept resurgence can be rare in some contexts; indeed, none of the representative images we sample reintroduce the unlearned concepts in the object erasure task. However, as both Figure 2 and Figure 4 demonstrate, concept resurgence can occur in striking and seemingly unpredictable ways, running the risk that developers or users can inadvertently reintroduce harmful or unwanted content.

---

[1]The GCD is a popular open source model for classifying celebrity images; Lu et al. (2024) document that the GCD achieves $> 99\%$ top-1 accuracy on celebrity images sampled from Stable Diffusion v1.4.

[2]The only exception is the Erase 5 Objects task, which we add to evaluate simultaneous erasure of multiple concepts.

[3]We say these are "representative" examples because we choose the *first* image generated for each prompt after fixing the random seed at 0, and use the same set of unlearned concepts, hyperparameters and prompts studied in Lu et al. (2024). Thus, these figures highlight the degree of concept resurgence in a "typical" case.

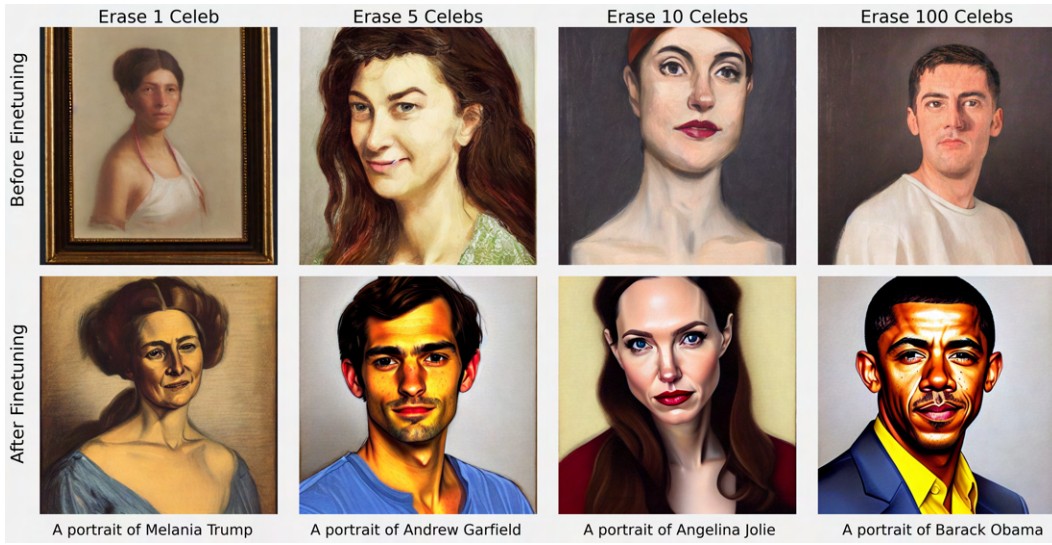

Figure 2: Representative images generated by SD v1.4 in each celebrity unlearning task before (top panel) and after (bottom panel) subsequent fine-tuning. Fine-tuning partially reintroduces the ability to generate the unlearned faces in the Erase 5, 10 and 100 celebrity tasks.

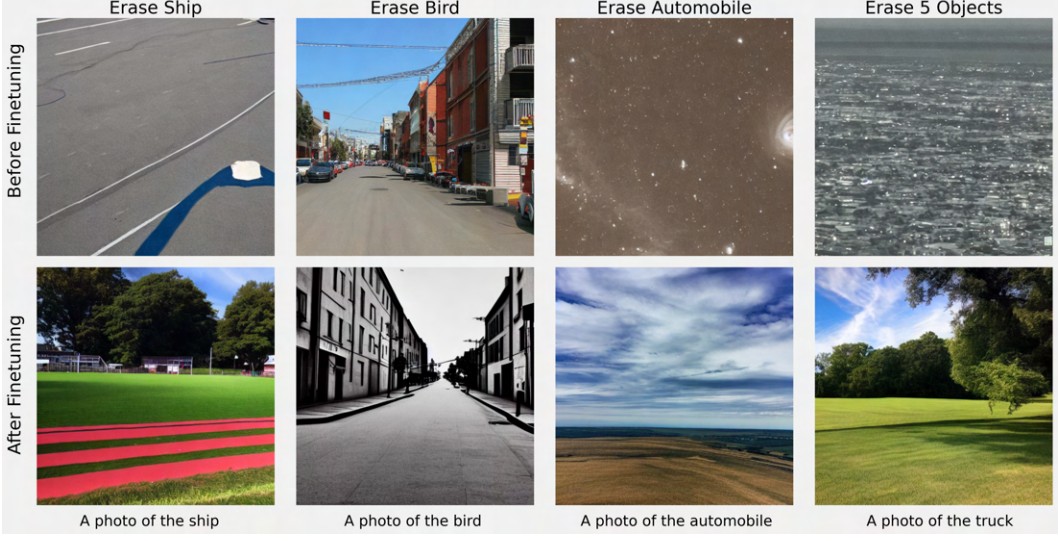

Figure 3: Representative images generated by SD v1.4 in each object unlearning task before (top panel) and after (bottom panel) subsequent fine-tuning. Unlike the celebrity erasure task, concept resurgence is rare, and none of the unlearned concepts are generated. However, as Figure 4 demonstrates, the vulnerability persists on certain concepts and prompts.

In Figure 5, we quantify the degree of resurgence across both the object and celebrity erasure tasks using the metrics described above. As suggested by the qualitative results, the degree of resurgence is substantially larger in the celebrity erasure task, particularly as the number of unlearned celebrities grows large (intuitively, such tasks are "harder" than unlearning a smaller number of celebrities, as the model must simultaneously unlearn many concepts without degrading model performance on other, unrelated tasks). The vulnerability appears in the object erasure task as well, albeit to a lesser degree. We emphasize however that in many contexts, even rare concept resurgence presents unacceptable risks. In the remainder of this work, we seek to systematically characterize *when* and *why* this resurgence occurs, focusing first on algorithmic choices (Section 4) and then on characteristics of the fine-tuning dataset (Section 5).

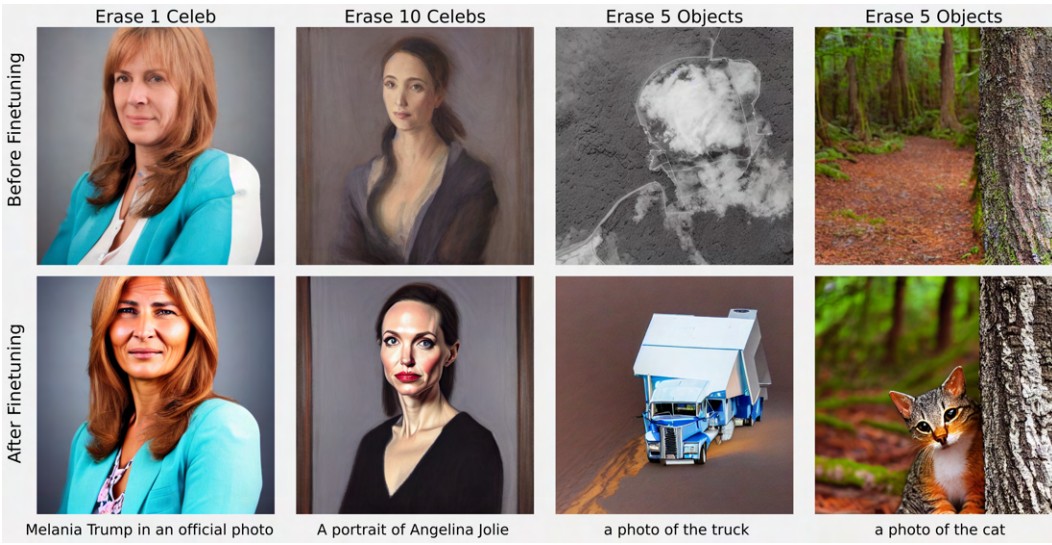

Figure 4: Selected images generated by SD v1.4 in both the celebrity and object unlearning tasks before (top panel) and after (bottom panel) fine-tuning. These examples demonstrate the risk of concept resurgence after an initial, seemingly successful, unlearning step.

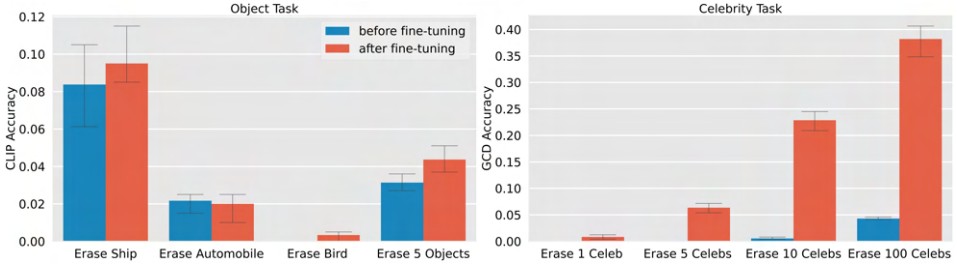

Figure 5: Quantifying the degree of concept resurgence in object and celebrity erasure. The celebrity erasure task demonstrates particularly severe resurgence as the number of unlearned celebrities grows large. Resurgence occurs to a more limited degree in the object erasure task.

## 4    ALGORITHMIC FACTORS DRIVING CONCEPT RESURGENCE

The compositional updating pipeline involves several algorithmic choices that can either contribute to or mitigate the risk of concept resurgence. We focus on common choices made across most concept unlearning algorithms for diffusion models, using MACE as our baseline technique to assess their impact. These choices include regularization in the cross-attention refinement update, mapping the embedding of the concept to be unlearned to more general concepts, and the selection of the algorithm used for fine-tuning. In the following two sections we focus our attention on the `erase 10 celebrities` and `erase ship` tasks as we vary other components of the training pipeline.

At a high level, we hypothesize that resurgence occurs because unlearning does not update the model parameters to be sufficiently "far away" from the pretrained weights. Thus, although unlearning may initially suppress the generation of unwanted concepts, even a modest degree of fine-tuning tends to shift model weights towards their initial state, and thus reintroduce seemingly erased concepts. To validate this hypothesis, we first assess the impact of three algorithmic factors on concept resurgence: regularization, choice of mapping concept, and the fine-tuning algorithm.

### 4.1    MAPPING CONCEPTS

First, we modify the mapping concepts used for each concept to be unlearned. This impacts both the cross-attention refinement objective and the final update applied to the model, which fuses multiple

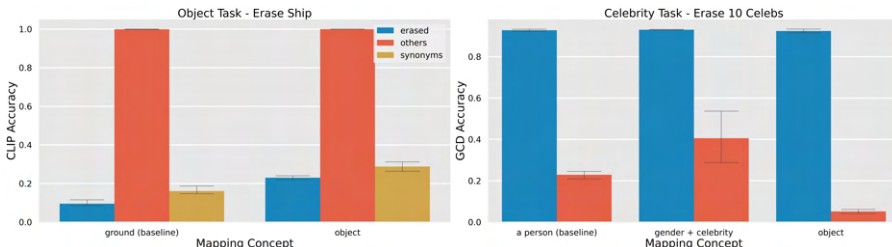

Figure 6: Effect of different generic mapping concepts on concept resurgence. More generic and unrelated concepts greatly reduce resurgence in both tasks, with near-total elimination in the celebrity erasure task when using objects as the mapping concept.

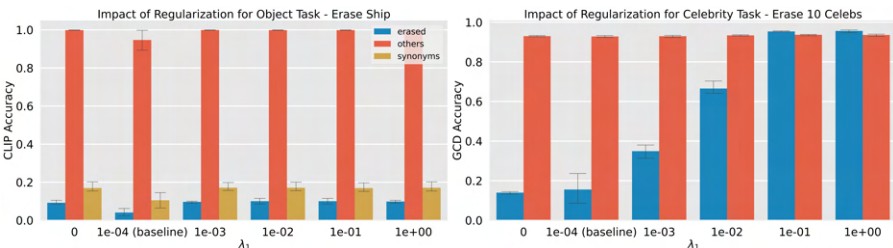

Figure 7: Impact of regularization in the cross-attention refinement weight update in both the object erasure (left) and celebrity erasure (right) tasks. Increasing regularization increases concept resurgence in the celebrity erasure task, but has little impact on the object erasure task.

LoRA modules corresponding to each unlearned concept. As described in Eq. (1), the embeddings of the words which co-occur with the concept to be unlearned are mapped to the corresponding embeddings for the mapping concept. To test our initial hypothesis, we select mapping concepts which are increasingly unrelated to the concept to be unlearned. This acts as a proxy for moving the initial model farther and farther away from its initial pretrained weights. For `erase 10 celebrities` we experiment with the following mapping concepts: "a person" (the baseline), "an object" and "a {male, female} celebrity," which correspond to more general and more specific concepts, respectively. For `erase ship` we experiment with "ground" (the baseline) and "object." We present these results in Figure 6.

As illustrated in Figure 6, concept resurgence is quite sensitive to the choice of mapping concept. In the object experiment, mapping "ship" to a different but *specific* concept ("ground") reduces resurgence more effectively than mapping it to a general concept like ("object"). In the celebrity erasure task, mapping each celebrity to a more specific concept (e.g., Jennifer Aniston → "a female celebrity") leads to more severe resurgence than if erased celebrities are mapped to more generic concepts. These findings are consistent with the hypothesis that while unlearning can initially suppress the generation of the unlearned concepts, it may do so through small changes in the parameter space, which can be easily undone by even modest degrees of further fine-tuning.

## 4.2 REGULARIZATION

A key algorithmic component in many approaches to concept unlearning in diffusion models, including MACE and Unified Concept Editing (UCE) (Gandikota et al., 2024) is modifying the cross-attention mechanism. This mechanism is responsible for encoding the prompt into an embedding that the diffusion process conditions on. In particular, MACE proposes to modify the cross-attention weights found via the solution to the following optimization problem:

$$\min_{\mathbf{W}'_k} \underbrace{\sum_{i=1}^{n} \|\mathbf{W}'_k \cdot \mathbf{e}_i^{\text{co}} - \mathbf{W}_k \mathbf{e}_i^{\text{gen}}\|_2^2}_{\substack{\text{guides embeddings of model} \\ \text{towards those of generic concept}}} + \lambda_1 \underbrace{\sum_{i=n+1}^{n+m} \|\mathbf{W}'_k \cdot \mathbf{e}_i^{\text{prior}} - \mathbf{W}_k \mathbf{e}_i^{\text{prior}}\|_2^2}_{\substack{\text{ensures embeddings of other concepts} \\ \text{remain accurate}}} , \qquad (1)$$

where $\mathbf{W}_k$ is the set of pretrained weights, $\mathbf{e}_i^{\text{co}}$ is the embedding of the $i^{\text{th}}$ word that co-exists in the prompt with the concept to be unlearned (e.g. the photo of {concept}), and $\mathbf{e}_i^{\text{gen}}$ is the embedding of the $i^{\text{th}}$ word that co-exists with the concept to be unlearned if that concept was replaced with its more generic concept (e.g. the photo of {generic concept}). Finally, $\mathbf{e}_i^{\text{prior}}$ is an embedding for a concept that we would like to preserve. Typically, the preserved concepts are generic ones that the model should still be able to generate after unlearning. The second term acts as a regularizer that keeps the new weights close to the original weights and thus prevent performance degradation on unrelated concepts.

We modify the regularization parameter $\lambda_1$ and assess its impact in the degree of concept resurgence. We run the MACE algorithm with $\lambda_1 \in \{0, .0001, .001, .01, .1, 1\}$ in the `erase 10 celebrities` and `erase ship` tasks. $\lambda_1 = 0$ assesses how much more robust the new parameters are to fine-tuning even if the overall performance of the model is much worse. At the other extreme, $\lambda_1 = 1$ allows us to assess how easy it is to reintroduce the unlearned concepts when the unlearning algorithm makes only small perturbations to the model parameters.

As Figure 7 demonstrates, the degree of regularization correlated with the degree of concept resurgence, particularly in the celebrity erasure task. Once the regularization parameter surpasses $\lambda = 1e\text{-}03$ we see that more than half of the images generated in the forget set are accurately classified as their concept. Indicating that larger updates to the initial pretrained weights are needed to ensure the prevention of concept resurgence. In the object task, we find that the regularization does not impact concept resurgence. Based on our results in Sec. 4.1, it is clear that the impact of mapping concepts on the phenomena of concept resurgence is first order. The object erasure task uses the mapping concept *ground* which is completely unrelated to ship. Celebrity erasure uses *person* as the mapping concept, which is much more related. Decreasing regularization helps prevent concept resurgence when the mapping concepts chosen are more related to the unlearned concepts.

## 4.3 FINE-TUNING ALGORITHM

Finally, we investigate the impact of fine-tuning the unlearned models with LoRA vs traditional full-parameter algorithms. While LoRA is the primary fine-tuning algorithm for diffusion models at the scale of Stable Diffusion we are interested in whether it helps prevent concept resurgence compared to full parameter finetuning. Since full parameter finetuning will have a much a larger update to all of the weights compared to LoRA.

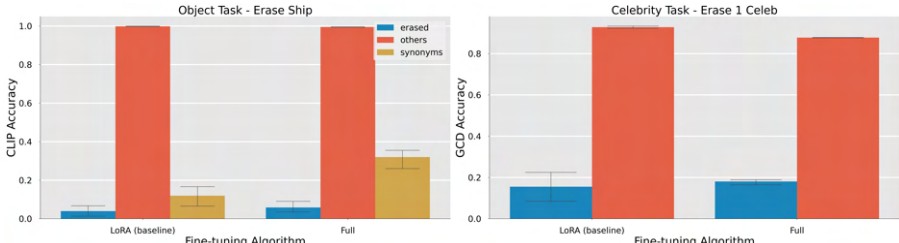

Figure 8: The choice of fine-tuning algorithm, whether using LoRA or full parameter fine-tuning, does not affect the extent of concept resurgence in either the object erasure tasks (left) or celebrity erasure tasks (right).

Fig. 8 demonstrates that the choice of finetuning algorithm does not help prevent concept resurgence. This supports our prior evidence that the mapping concepts and regularization are two core compo-

nents contributing to concept resurgence. Suggesting our hypothesis that larger updates are required by the unlearning algorithm to prevent concept resurgence is one reason for this phenomenon.

## 5 DATA-DEPENDENT FACTORS DRIVING CONCEPT RESURGENCE

The second component of the pipeline we investigate is the data the unlearned models are fine-tuned on. As discussed in Section 3, we choose not to focus on adversarial dataset constructions (including e.g., simply directly fine-tuning a model on concepts which were previously unlearned); for work on attacking or jailbreaking text-to-image diffusion models we instead refer to Ma et al. (2024); Yang et al. (2024); Dong et al. (2024). Instead, we consider the kind of dataset constructions that occur as part of common and benign use. For example, the end user of an open source diffusion model may want to fine-tune the model to acquire new concepts which were excluded from the pretraining set and/or improve its performance on particular tasks of interest. In Section 3, we demonstrated that fine-tuning on random in-domain concepts (e.g., fine-tuning on randomly chosen celebrities after unlearning) can lead to concept resurgence. In this section, we seek to further investigate the role of dataset construction in concept resurgence. First, we consider fine-tuning on in-domain concepts with varying levels of similarity to the concepts which are initially unlearned. Intuitively, it may be that fine-tuning on more related concepts can exacerbate the degree of concept resurgence. We then turn to fine-tuning on *out-of-domain* concepts; e.g., fine-tuning on images of randomly chosen objects after a celebrity unlearning task. We describe these experiments in more detail below.

### 5.1 CLIP DISTANCE

First, we consider fine-tuning on unrelated but in-domain concepts as described above. We further segment these concepts by thresholding the CLIP distance to the unlearned concepts, which we use as a proxy for how "related" the fine-tuning dataset is to the unlearned concepts. In particular, for each task we find a publicly available dataset with hundreds of concepts in the same domain (e.g., for the celebrity unlearning task, this is a dataset of celebrity images). We describe these datasets in detail in Appendix C.2. For each concept in the corresponding dataset, we compute its maximum CLIP cosine similarity over the set of unlearned concepts. We then partition this fine-tuning set into three evenly sized subsets based on the terciles of the CLIP cosine similarity. Finally, we randomly sample 10 concepts from each tercile to create different fine-tuning datasets which vary in their degree of "relatedness" to the unlearned concepts. We present these results in Figure 9.

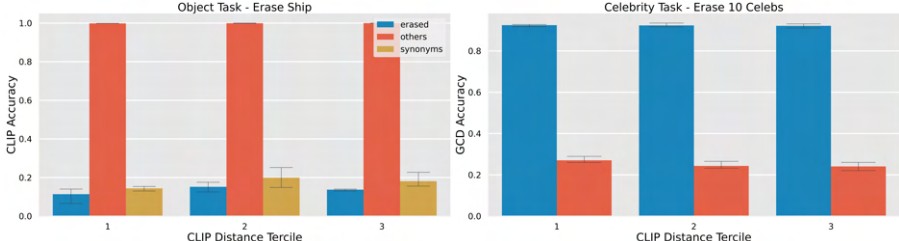

Figure 9: The degree of concept resurgence remains constant in both tasks, object (left) and celebrity (right) erasure regardless of the similarity (as measured by CLIP distance) of the fine-tuning dataset to the unlearned concepts.

As Figure 9 demonstrates, the degree to which the fine-tuning set is related to the unlearned concepts — at least as measured by CLIP cosine similarity — does not appear to meaningfully correlate with the degree of concept resurgence. This finding underscores the danger of concept resurgence even when fine-tuning on relatively unrelated data.

### 5.2 OUT OF DOMAIN CONCEPTS

Finally, to better understand the scope of concept resurgence, we curate an additional set of fine-tuning datasets which contain "out-of-domain" concepts which are wholly unrelated to those in the unlearning task. In particular, for each unlearning task, we fine-tune the resulting model on the

random concept datasets from the other two unlearning tasks (e.g., for the celebrity unlearning task we further fine-tune the resulting model on the same randomly selected objects used as the initial fine-tuning set in the object erasure task).

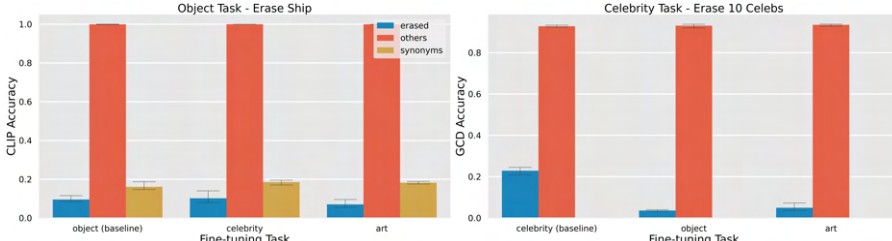

Figure 10: Concept resurgence is much less prevalent when finetuning on out of domain concepts. In both the object (left) and celebrity (right) tasks the degree of concept resurgence reduces when fine-tuning on out of domain concepts compared to in domain ones.

As Figure 10 demonstrates, the degree of concept resurgence does vary with the domain of the fine-tuning set, particularly for the celebrity task. In particular, fine-tuning on in domain images appears to exacerbate the risk of concept resurgence, while fine-tuning on out of domain images can mitigate it. Intuitively, this suggests that fine-tuning on images which are wholly unrelated to the unlearned concepts can be safer than fine-tuning on more closely related images, and lends support to the hypothesis laid out in Section 4.

## 6 DISCUSSION AND LIMITATIONS

The scale of generative models introduces new challenges, including the risk of learning concepts that are unsuitable or undesirable for certain downstream applications. Ideally, unlearning methods would allow model developers to precisely and permanently remove unwanted concepts while preserving the model's overall utility. Unfortunately, reality is more complex.

Our work uncovers a critical limitation of current unlearning methods, which we termed *concept resurgence*. We demonstrate this phenomenon through rigorous empirical evaluations, highlighting the practical limitations of state-of-the-art unlearning techniques. These findings emphasize the need to rethink current approaches to concept erasure, especially in contexts where maintaining the integrity of model updates is essential.

Our investigation opens up several important avenues for future work. For example, we do not provide a theoretical characterization of concept resurgence, nor do we present a strategy designed to prevent it from happening. Both developments could help to enhance the robustness of unlearning methods. Additionally, though our evaluations focus on well-known academic benchmarks, further research is necessary to assess the prevalence of concept resurgence in real-world deployments.

Concept resurgence also raises important questions about responsibility for downstream harms. Despite a developer's best efforts to sanitize a model using these techniques, a downstream user who fine-tunes a published model might be surprised to discover that guardrails put in place by the developer no longer exist. This creates a dilemma: is the developer obligated to permanently and irrevocably erase problematic concepts, or does responsibility shift to the downstream when they (inadvertently) reintroduce them?

Despite these challenges, concept unlearning remains a valuable tool for model developers. By identifying its vulnerabilities, our work aims to drive the development of erasure techniques that remain robust throughout a model's life-cycle or develop tools that can help developers anticipate when concept resurgence is likely to happen. Addressing these weaknesses will be essential for ensuring the safety and alignment of generative models as they are fine-tuned and adapted for diverse applications.

## 7 ETHICS STATEMENT

This research focuses on evaluating the robustness of concept unlearning in diffusion models to downstream fine-tuning. As discussed in Appendix A, we strictly avoid explicit content generation in this work. Our work exposes a novel vulnerability in concept unlearning for text-to-image diffusion models, and thus, as is common for security or AI alignment research, risks misuse by malicious actors to bypass safety training and reintroduce unwanted or harmful behavior. Nonetheless, consistent with the ethos and best practices in these fields, we believe that openly discussing and reporting vulnerabilities is crucial to the development of safe and reliable generative models. Regardless of whether we report it, the vulnerability exists. Thus, it is beneficial to identify and report it now so that current users are aware and future researchers can help to mitigate it.

## 8 REPRODUCIBILITY STATEMENT

Our manuscript includes all materials needed to reproduce our experimental results from scratch. In particular, the supplementary material includes code, data and detailed instructions to run our experimental pipeline, including low-level configuration choices and random seeds. Where relevant, all additional detail regarding our experimental setup (e.g., hyperparameters, evaluation prompts, erased and retained concepts and the precise dataset curation procedures) are detailed in Section 3, Appendix C.1 and Appendix C.2. Finally, because much of our work is intended to provide a fair comparison to Lu et al. (2024), which is the current state of the art in concept unlearning, all other choices are exactly as defined in their publicly available Github repository unless explicitly specified in our work.

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

## A    EXCLUDING ARTISTIC STYLE AND EXPLICIT CONTENT ERASURE

As discussed in Section 3, celebrity and object erasure are two of the four benchmarks considered in Lu et al. (2024). The other two considered are artistic style erasure — unlearning the ability to generate images in the style of specific artists, e.g., due to copyright concerns — and explicit content erasure, particularly to suppress nudity. We exclude artistic style erasure due to the difficulty of quantifying the *degree* of a particular style in an image. In particular, unlike celebrity or object erasure, artistic style is not localized to specific regions in an image, and instead is a holistic (and partially subjective) property of the model output. For example, Drouillard et al. (2024) note that "courts have emphasized the importance of considering the 'total concept and overall feel' [for determining whether copyright infringement has occurred], rather than relying on mechanical dissection or quantitative measures alone." Characterizing artistic style replication (and copyright infringement more broadly) is rich topic in its own right, and we refer to Tenenbaum & Freeman (1996); Somepalli et al. (2024); Zhang et al. (2023b); Casper et al. (2023) for additional background.

We further exclude the explicit content benchmark due to the sensitive and unpredictable nature of the images which may be generated by the model, the lack of agreed upon standards for conducting such evaluations responsibly, and recent well publicized examples of the real-world harm that can result from synthetic but realistic nude images (Lapowsky, 2023; Grose, 2024). Instead, we use the object and celebrity erasure tasks as representative but benign benchmarks on which to conduct our evaluations.

## B    IMPACT OF FINE-TUNING ON RETAINED CONCEPTS

Below we first examine the analogue of Figure 5 on the retained set, which is presented in Figure 11. Consistent with Lu et al. (2024), we find that MACE preserves model performance on the set of retained concepts, and furthermore, subsequent fine-tuning does not degrade performance on this set.

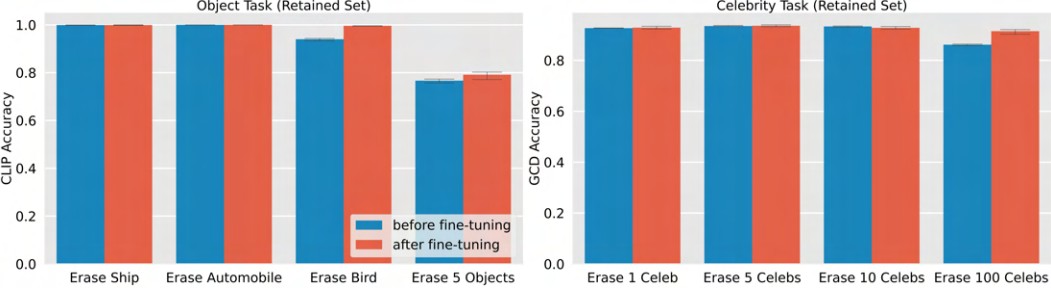

Figure 11: The performance on the retain set remains almost constant before and after finetuning in both tasks. It slightly increases when the number of concepts being erased is at its largest in both tasks (i.e. when erasing 5 objects and 100 celebrities).

As an additional sense check, we examine the CLIP and FID scores on random objects sampled from COCO-5K (as described in Section 3) before and after fine-tuning in the celebrity erasure tasks. These are presented in Figure 12 and Figure 13, respectively. We see that the CLIP scores remain almost identical, while the FID scores increase (i.e., degrade) after fine-tuning. The results of these three figures are thus broadly consistent with fine-tuning not degrading performance across a variety of tasks; if anything, concept resurgence can occur even if overall performance (i.e., on unrelated tasks) *decreases* slightly.

## C    FINE-TUNING DATASET CURATION

### C.1    RANDOM

In this section we provide additional details related to the dataset curation process for the different tasks. The "random" dataset for celebrities, includes 25 images of 10 distinct celebrities, chosen

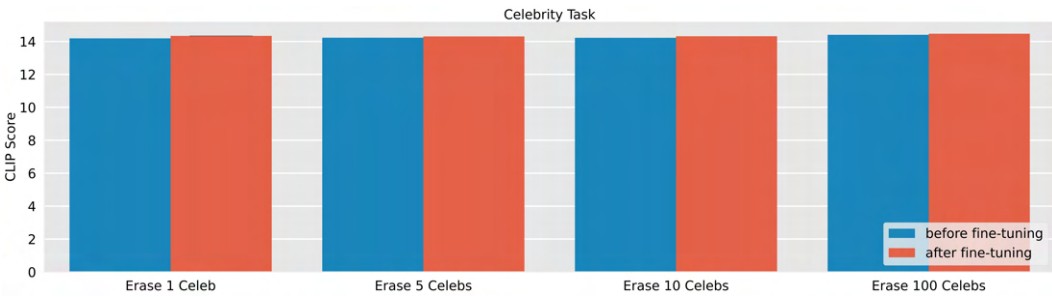

Figure 12: The CLIP score on unrelated objects sampled from COCO-5K remains almost constant before and after finetuning in the celebrity erasure task.

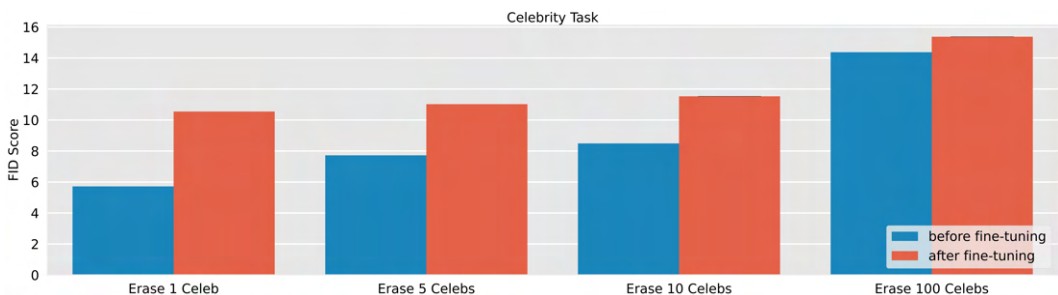

Figure 13: The FID score on unrelated objects sampled from COCO-5K degrades slightly after fine-tuning in the celebrity erasure task.

arbitrarily from those used in Lu et al. (2024) while ensuring that they do not overlap with any of the erased celebrities in any of the subtasks. These celebrities are Amy Winehouse, Elizabeth Taylor, George Takei, Henry Cavill, Jeff Bridges, Jensen Ackles, Jimmy Carter, Kaley Cuoco, Kate Upton and Kristen Stewart. For each celebrity, we generated five images for each of five prompts (25 total). These prompts were:

1. "A portrait of [name]"
2. "An image capturing [name] at a public event"
3. "A sketch of [name]"
4. "An oil painting of [name]"
5. "[name] in an official photo"

The "random" dataset for objects, includes 5 images of 8 distinct objects, chosen arbitrarily from the classes of CIFAR-100 Krizhevsky et al. (2009) while ensuring that they do not overlap with any of the erased objects. These objects are trout, ray, bee, rose, lobster, girl, oak tree, aquarium fish, Kate Upton and Kristen Stewart. For each object, we generated five images for each prompt. The prompt used was "a photo of the [object]."

## C.2    CLIP DISTANCE

In this section we provide additional details related to the dataset curation process for Sec. 5 for the different tasks. For celebrities, we start with all of the celebrities from the CelebA-HQ-Dialog (Karras et al., 2018; Jiang et al., 2021) dataset. We compute the CLIP embeddings of the 10177 celebrities from this dataset and the 10 celebrities being unlearned. Using these CLIP embeddings we compute the cosine similarity between every unlearned celebrity and the 10177 celebrities in CelebA-HQ-Dialog. We find the minimum and maximum similarity to be 0.17 and 0.80 respectively. We then construct terciles in this interval based on the minimum similarity between the celebrity in CelebA-HQ-Dialog and the unlearned celebrity, ensuring that at least 10 of the celebrities in CelebA-HQ-Dialog fall into each tercile. We then sample 10 celebrities from each tercile and generate a fine-tuning dataset with those celebrities in the same way as the random dataset.

| Tercile | Cosine Similarity Interval | Celebrities |
|---|---|---|
| 1 | 0.17 - 0.37 | Elize Du Toit, Heather Marie Mardsen, Soleil Moon Frye, Eniko Mihalik, Mia Wasikowska, Ruslaan Mumtaz, Petra Cubonova, Karin Dor, Kathyrn Erbe, Justine Mattera |
| 2 | 0.37 - 0.58 | Delta Goodrem, Babs Jongh, Tom Green, Melissa Haro, Ratan Tata, Danielle Darrieux, Eike Batista, Johnny Borrell, Scott Stiner, Amy Davidson |
| 3 | 0.58 - 0.80 | Tamara Ecclestone, Bryan Cranston, Gregg Sulkin, Sigrid Agren, Ty Pennington, Noemie Lenoir, Jana Ina, Jonathan Tucker, Valerie Bertinelli |

Table 1: Celebrity concepts used in each of the fine-tuning datasets for the CLIP distance experiments.

For objects, we start with all of the artists from the CIFAR100 (Krizhevsky et al., 2009) dataset. We compute the CLIP embeddings of the 100 objects from this dataset and the 5 objects being unlearned. Using these CLIP embeddings we compute the cosine similarity between every unlearned object and the 100 artists in CIFAR100. We find the minimum and maximum similarity to be 0.68 and 0.84 respectively. We then construct terciles in this interval based on the minimum distance between the object in CIFAR100 and the 5 unlearned objects, ensuring that at least 10 of the objects in CIFAR100 fall into each tercile. We then sample 10 objects from each tercile and generate a fine-tuning dataset with those objects in the same way as the random dataset.

| Tercile | Cosine Similarity Interval | Objects |
|---|---|---|
| 1 | 0.68 - 0.74 | tulip, plain, bowl, pine tree, mountain, house, crab, willow tree, motorcycle, mushroom |
| 2 | 0.75 - 0.79 | streetcar, maple tree, seal, orange, cup, flatfish, sunflower, shark, hamster, aquarium fish |
| 3 | 0.80 - 0.83 | tiger, tank, turtle, cloud, orchid, road, elephant, rocket, bee, raccoon |

Table 2: Object concepts used in each of the fine-tuning datasets for the CLIP distance experiments.

