# OpenReview forum: "Unstable Unlearning: The Hidden Risk of Concept Resurgence in Diffusion Models"
_ICLR.cc/2025/Conference — ICLR 2025 Conference Withdrawn Submission_

### Official Review · Reviewer_D6Um · 2024-10-28

**Soundness:** 2
**Presentation:** 3
**Contribution:** 2
**Rating:** 5
**Confidence:** 4

**Summary:**

The paper investigates the phenomenon of concept resurgence in text-to-image diffusion models that have been fine-tuned to forget certain concepts. The authors show that after erasing certain concepts with MACE, fine-tuning on unrelated concepts can reintroduce the erased concepts. The authors carry out experiments where several parameters of the erasing/fine-tuning are varied to elucidate the various factors that contribute to concept resurgence.

**Strengths:**

- This idea of concept resurgence is very interesting and pertinent to the safety/concept unlearning community in text-to-image models. To my knowledge this is the first work to identify such an issue.
- The paper is well-written and ideas are clearly communicated.

**Weaknesses:**

- The experiments in the paper are only on models erased with MACE, although numerous SD erasure works [1,2] have been proposed. Without experiments on a few more baselines, logically speaking the evidence from the paper only supports the claim that concept resurgence occurs on models erased with MACE rather than in general, which would weaken its impact.

- Sec 4.3 seems to contradict the hypothesis that concept resurgence is more prominent if the weights from erasure were not moved far from the original weights, since I would assume LoRA makes smaller weight changes than full fine-tuning, yet the effects on resurgence are similar. Could the authors make this more quantitative and measure the deviation of the weights from the original values, for e.g., in the L2 sense?

- Have the authors tested resurgence on truly more 'abstract' concepts like nudity or violence? The current experiments focus on relatively 'easier' concepts that can be defined by a single or few synonyms, like the name of the celebrity or object. Concepts like nudity can be expressed by numerous synonyms and even abstractly by the names of artists who paint with nude styles, for example.

- Overall I found that the technical contribution of the paper to be somewhat lacking by ICLR's standards, even though the phenomena presented is novel. The experiments are focused on one baseline and two concept types (celebrities and objects). As the authors acknowledge in the limitations, the paper lacks theoretical insights into concept resurgence or any mitigation strategies.

Minor points:
- consider moving Eq 1 to the front of the paper and introduce MACE more thoroughly given that the experiments in the paper are focused on MACE.
- some missing references on early works in the area of erasure/safety in text-to-image models [1,2,3,4].

[1] Zhang, Eric, et al. "Forget-Me-Not: Learning to Forget in Text-to-Image Diffusion Models. ArXiv abs/2303.17591 (2023)." (2023).

[2] Gandikota, Rohit, et al. "Erasing concepts from diffusion models." Proceedings of the IEEE/CVF International Conference on Computer Vision. 2023.

[3] Heng, Alvin, and Harold Soh. "Selective amnesia: A continual learning approach to forgetting in deep generative models." Advances in Neural Information Processing Systems 36 (2024).

[4] Schramowski, Patrick, et al. "Safe latent diffusion: Mitigating inappropriate degeneration in diffusion models." Proceedings of the IEEE/CVF Conference on Computer Vision and Pattern Recognition. 2023.

**Questions:**

- Can the authors provide details on what the 'others' and 'synonyms' are in the different figures?
- Could the authors provide more experiment details, for e.g., what was the fine-tuning procedure to induce concept resurgence? Information like hyperpameters to reproduce the experiments are missing

---

### Official Review · Reviewer_vhR8 · 2024-11-04

**Soundness:** 2
**Presentation:** 2
**Contribution:** 3
**Rating:** 5
**Confidence:** 3

**Summary:**

This paper reports an interesting behavior of unlearned diffusion models, called concept resurgence – when a concept is unlearned from a diffusion model, this concept is observable again after fine-tuning. The cause of this phenomenon is analyzed in two ways: algorithmic factors and data-dependent factors. In short, concept resurgence occurs when unlearned model parameters are close to the parameters of a pre-trained model and when fine-tuning data is correlated to training sets.

**Strengths:**

This paper introduces a very interesting phenomenon of unlearned models – concept resurgence. To my understanding, this observation hasn’t been discussed in the unlearning domain.

**Weaknesses:**

The supporting experiments are slightly below the ICLR standard. Overall, this paper should justify their claim via experiments but the supporting experiments are weak/handful.

* The interesting phenomenon is only evaluated on one unlearning model (i.e., MACE). Additional unlearning methods are needs to be evaluated on hopefully five different methods, e.g., Selective Amnesia (https://arxiv.org/abs/2305.10120), SALIENCY UNLEARNING (https://arxiv.org/abs/2310.12508), and more.
* For out-of-domain concepts, it would be useful to add some visualization on the correlation between a target concept and out-of-domain concepts.

**Questions:**

* Can you apply additional unlearning methods (hopefully five different methods) to show the same concept resurgence phenomenon?
* Can you visualize a target concept and out-of-domain concepts to show some (semantic) distance between them?

---

### Official Review · Reviewer_G92E · 2024-11-04

**Soundness:** 1
**Presentation:** 3
**Contribution:** 3
**Rating:** 3
**Confidence:** 4

**Summary:**

This paper focuses on the concept of “concept resurgence” in text-to-image diffusion models. These models are often updated incrementally through fine-tuning and unlearning steps. The authors demonstrate that fine-tuning a diffusion model can cause previously “unlearned” concepts to reappear, even when fine-tuning on seemingly unrelated data. They conduct experiments using Stable Diffusion v1.4 and the Mass Concept Erasure (MACE) technique. The study investigates factors contributing to concept resurgence, including algorithmic choices (such as mapping concepts, regularization, and fine-tuning algorithms) and data-dependent factors (like CLIP distance and out-of-domain concepts). The findings highlight the fragility of current model update paradigms and raise concerns about ensuring the safety and alignment of diffusion models.

**Strengths:**

* The authors identify a previously unknown vulnerability (concept resurgence) in diffusion models, which is important for understanding the limitations of current model update strategies.
* This paper systematically examines both algorithmic and data-dependent factors contributing to concept resurgence, providing a detailed understanding of the phenomenon.
* The research has direct implications for the development and safety of diffusion models, as it highlights the need to address concept resurgence to ensure reliable and safe model performance.

**Weaknesses:**

* While this paper focuses mainly on MACE as the unlearning algorithm, it remains unclear whether the observed results could be fully generalizable to other unlearning techniques, which can be considered to add for more comprehensive analysis.
* Since we cannot enumerate all possible concepts during evaluation, could the authors provide some insights on the metrics that we can use to measure the difficulty of the resurgence of a certain concept? This might help to reach a more general conclusion of the experiments.
* Aside from the two examined celebrity and object erasure tasks and specific benchmarks, it would be better to extend the evaluation on more diverse settings to see if the findings still hold.
* Minor: Though it might be out of the scope of this manuscript, it is very interesting to have some theoretical analysis regarding the observations.

**Questions:**

Please kindly refer to the Weaknesses.

---

### Official Review · Reviewer_CbHp · 2024-11-05

**Soundness:** 2
**Presentation:** 2
**Contribution:** 2
**Rating:** 3
**Confidence:** 4

**Summary:**

The paper examines a significant vulnerability in text-to-image diffusion models regarding the unlearning of unwanted concepts, termed "concept resurgence." It demonstrates that fine-tuning diffusion models on seemingly unrelated and benign data can inadvertently lead to the re-emergence of previously erased concepts. This vulnerability raises serious concerns about the reliability of current unlearning methods, particularly for developers aiming to protect users from undesirable content. The authors conducted experiments using Stable Diffusion v1.4 and the Mass Concept Erasure (MACE) technique, revealing that concept resurgence can occur even under benign conditions. Further, the authors explore and try to identify  various factors which may contribute to this issue such as the choice of fine-tuning data and the regularization applied during unlearning.

**Strengths:**

- The paper tackles a timely and practically-relevant problem supported by a fair amount of experiments. Model unlearning regarding AI safety is an area with limited prior research, making this work particularly valuable.
- This work stands as a pioneering study in attempting to identify concept resurgence phenomenon regarding text-to-image diffusion models.

**Weaknesses:**

- The main weakness of this paper is its limited experimental scope. While the paper's key contribution is the concept resurgence phenomenon, it is supported only by limited empirical evidence. This calls for testing the phenomenon in various setups, yet the authors only use a single model, SD v1.4. Given the availability of advanced models such as SDXL, EDM, MDT, and FLUX, it would be helpful to see experiments using other diffusion models, particularly those trained with flow matching objectives instead of score matching losses. Additionally, the authors exclude tasks related to artistic style removal and explicit content removal, citing evaluation challenges. However, it would still be valuable to demonstrate the concept resurgence phenomenon in these tasks, even if a fair evaluation is difficult. The current experimental setup is also limited in terms of dataset diversity. Providing additional qualitative examples beyond Figures 2 and 4 would strengthen the paper.
- To my understanding, this paper only experimented with a single unlearning technique, MACE. The authors need to explore more existing methods such as UCE, FMN, ESD, SDD etc. Even if MACE is a SOTA unlearning method, concept resurgence may not appear with the other baselines. Section 4.2, in particular, would benefit from a broader discussion of baseline methods.
- The authors propose three potential contributors to concept resurgence: mapping concept, regularization, and fine-tuning algorithms. However, the discussion in Section 4 lacks depth. The authors should offer theoretical justifications or at least propose a main hypothesis supported by empirical evidence. For example, in Figure 7, they suggest that “increasing regularization increases concept resurgence in the celebrity erasure task, but has little impact on the object erasure task.” It would be helpful to identify the key factor causing this difference and explore how this factor might be used to prevent concept resurgence.  Further, the authors conclude that the difference between full fine-tuning and LoRA fine-tuning does not affect concept resurgence. However, if sufficiently “distant” fine-tuning can prevent concept resurgence, wouldn’t full fine-tuning be more effective than LoRA in doing so?

**Questions:**

- What exactly is meant by “mapping concept”? I read the paper carefully but still find the term’s exact definition unclear. Did the authors use this term in the same way as in the MACE paper?
- Regarding Figure 5, what would happen if 10 or 100 objects were removed, as in the celebrity erasure task?

---

### Author Response · Authors · 2024-11-21

We thank the reviewers for their thoughtful and constructive comments. Unfortunately, we cannot fully incorporate their feedback during the rebuttal period, and have thus opted to withdraw our work from ICLR. We look forward to strengthening our manuscript and resubmitting at a later date.

---

### Note · Authors · 2024-11-21

**Comment:**

We thank the reviewers for their thoughtful and constructive comments. Unfortunately, we cannot fully incorporate their feedback during the rebuttal period, and have thus opted to withdraw our work from ICLR. We look forward to strengthening our manuscript and resubmitting at a later date.

**Withdrawal Confirmation:**

I have read and agree with the venue's withdrawal policy on behalf of myself and my co-authors.